# Population ageing and mortality during 1990–2017: A global decomposition analysis

Xunjie Cheng[1], Yang Yang[2,3], David C. Schwebel[4], Zuyun Liu[5], Li Li[6], Peixia Cheng[1], Peishan Ning[1], Guoqing Hu[1,7]*

1 Department of Epidemiology and Health Statistics, Xiangya School of Public Health, Central South University, Changsha, China, 2 Department of Biostatistics, College of Public Health and Health Professions, University of Florida, Gainesville, Florida, United States of America, 3 Emerging Pathogens Institute, University of Florida, Gainesville, Florida, United States of America, 4 Department of Psychology, University of Alabama at Birmingham, Birmingham, Alabama, United States of America, 5 Department of Pathology, Yale School of Medicine, New Haven, Connecticut, United States of America, 6 Division of Epidemiology, College of Public Health, Ohio State University, Columbus, Ohio, United States of America, 7 National Clinical Research Center for Geriatric Disorders, Xiangya Hospital, Central South University, Changsha, China

* huguoqing009@gmail.com

## Abstract

### Background

As the number of older people globally increases, health systems need to be reformed to meet the growing need for medical resources. A few previous studies reported varying health impacts of population ageing, but they focused only on limited countries and diseases. We comprehensively quantify the impact of population ageing on mortality for 195 countries/territories and 169 causes of death.

### Methods and findings

Using data from the Global Burden of Disease Study 2017 (GBD 2017), this study derived the total number of deaths and population size for each year from 1990 to 2017. A decomposition method was used to attribute changes in total deaths to population growth, population ageing, and mortality change between 1990 and each subsequent year from 1991 through 2017, for 195 countries/territories and for countries grouped by World Bank economic development level. For countries with increases in deaths related to population ageing, we calculated the ratio of deaths attributed to mortality change to those attributed to population ageing. The proportion of people aged 65 years and older increased globally from 6.1% to 8.8%, and the number of global deaths increased by 9 million, between 1990 and 2017. Compared to 1990, 12 million additional global deaths in 2017 were associated with population ageing, corresponding to 27.9% of total global deaths. Population ageing was associated with increases in deaths in high-, upper-middle-, and lower-middle-income countries but not in low-income countries. The proportions of deaths attributed to population ageing in 195 countries/territories ranged from −43.9% to 117.4% for males and −30.1% to 153.5% for females. The 2 largest contributions of population ageing to disease-specific deaths globally between 1990 and 2017 were for ischemic heart disease (3.2 million) and stroke

**Data Availability Statement:** All data used in this research were derived from the online resources of the Global Burden of Disease Study 2017 (http://

ghdx.healthdata.org/gbd-results-tool; http://ghdx.
healthdata.org/gbd-2017).

**Funding:** The authors received no specific funding
for this work.

**Competing interests:** The authors have declared
that no competing interests exist.

**Abbreviations:** GBD 2017, Global Burden of
Disease Study 2017.

(2.2 million). Population ageing was related to increases in deaths in 152 countries for
males and 159 countries for females, and decreases in deaths in 43 countries for males and
36 countries for females, between 1990 and 2017. The decreases in deaths attributed to
mortality change from 1990 to 2017 were more than the increases in deaths related to popu-
lation ageing for the whole world, as well as in 55.3% (84/152) of countries for males and
47.8% (76/159) of countries for females where population ageing was associated with
increased death burden. As the GBD 2017 does not provide variances in the estimated
death numbers, we were not able to quantify uncertainty in our attribution estimates.

## Conclusions

In this study, we found that population ageing was associated with substantial changes in
numbers of deaths between 1990 and 2017, but the attributed proportion of deaths varied
widely across country income levels, countries, and causes of death. Specific preventive
and therapeutic techniques should be implemented in different countries and territories to
address the growing health needs related to population ageing, especially targeting the dis-
eases associated with the largest increase in number of deaths in the elderly.

## Author summary

### Why was this study done?

- Evidence on the change in number deaths related to population ageing is important for
  each individual government to improve its healthcare system to address the increasing
  healthcare needs of older adults.

- Previous research assessing changes in health indicators (e.g., number of deaths, mortal-
  ity) influenced by population ageing was limited to specific countries or specific
  diseases.

- Quantitative methods for decomposing changes in the total number of deaths that were
  adopted by previous studies are sensitive to the choice of the decomposition order of the
  3 factors—population growth population ageing, and age-specific mortality rate—and
  the selection of reference group.

### What did the researchers do and find?

- Using a decomposition method that is not influenced by the selection of decomposition
  order of the 3 factors and the choice of the reference group, we conducted a comprehen-
  sive analysis to quantify the impact of population ageing on changes in the number of
  deaths in 195 countries/territories, and for 169 causes of death, from 1990 to 2017.

- Changes in the number of deaths related to population ageing varied greatly across the
  195 countries/territories; the attributed proportion ranged from −43.9% to 117.4% for
  males and −30.1% to 153.5% for females.

- The causes of death for which population ageing was associated with the greatest increases in global deaths between 1990 and 2017 were ischemic heart disease (3.2 million) and stroke (2.2 million).

- The decreases in deaths attributed to mortality change exceeded the increases in deaths related to population ageing between 1990 and 2017 for the whole world, as well as in 55.3% (84/152) of countries for males and 47.8% (76/159) of countries for females where population ageing was associated with increased death burden.

### What do these findings mean?

- Globally, population ageing was related to increases in deaths, highlighting the importance and urgency of improving health systems to meet the health needs of older adults.

- Varying death burden related to population ageing suggests flexible health policies should target the leading causes of attributed death burden in different countries/ territories.

- The death burden related to population ageing could be alleviated or even overcome through implementation of evidence-based interventions to reduce mortality.

## Introduction

Largely as a result of socioeconomic development, the global population has aged rapidly in the last few decades. The ageing population imposes a growing disease burden on the healthcare systems of the world, especially to prevent and treat certain types of diseases and injuries. According to a United Nations report [1], the number of people aged 65 years and older is expected to rise from 0.7 billion (9%) worldwide in 2019 to 1.5 billion (16%) in 2050. Another recent report suggested population ageing will be associated with a 55% increase in global disability-adjusted life years (DALYs) among people aged 60 years and older between 2004 and 2030 [2], indicating more medical resources will be needed to meet the healthcare needs of the elderly worldwide. Health systems in many nations will require reforms to meet this demand based on the health impact of population ageing.

Previous studies have explored specific aspects of the health impact of population ageing and provided policy-makers and researchers with some valuable information [3–8]. As an example, Moran et al. projected that cardiovascular events will increase by more than 50% between 2010 and 2030 in China as the result of population ageing and population growth [8]. However, most studies either focused on selected geographical locations (e.g., United States [9,10] or England and Wales [11,12]), making their conclusions not generalizable, or focused on selected diseases (e.g., coronary heart disease [8,11,13] or cancer [4,14]).Some studies did not separate the effects of population ageing (typically approximated as changes in age structure [3,4]) from those of population growth [8]. Without separating these effects, accurate estimation of the net effect of population ageing cannot be assessed, and results can be misleading. In addition, studies that estimated the net effect of population ageing adopted different decomposition methods [4,15–17]. These traditional methods are sensitive to the decomposition order of the 3 components (population growth, population ageing, and mortality change) as

well as to the choice of reference group [18], leading to inconsistent results across studies even using the same data. Finally, some studies relied on questionable assumptions, such as stability of age-specific mortality rates or incidence rates in the future [6,8].

To our knowledge, systematic analyses of the health impact of global population ageing across a long time period are absent in the published literature, restricting international organizations such as the World Health Organization and individual governments from making data-driven modifications of their healthcare systems to address the increasing health needs of the senior population. We recently developed a decomposition method that is not influenced by decomposition order and choice of reference group [18]. We used this method to estimate the impact of population ageing on global deaths; to assess the impact by sex, cause of death, and country; and to assess how changes in mortality rates affected the impact of population ageing from 1990 to 2017 globally and nationally.

## Methods

### Data source

All data were derived from online resources of the Global Burden of Disease Study 2017 (GBD 2017) [19,20]. As detailed elsewhere [15], GBD 2017 used 7 types of data sources to estimate numbers of deaths by age, sex, and cause of death for 195 countries and territories. Multiple strategies were adopted to impute missing data and to correct underreporting and misclassification, including (a) reattribution of deaths with garbage codes based on the method established by Ahern et al. [21], (b) disaggregation of causes of death that are condensed into aggregated groups according to the invariant relative risks of death by age group compared to a reference age group [15], and (c) noise reduction of 0 counts using a Bayesian noise reduction algorithm [15]. GBD 2017 also estimated age- and sex-specific population sizes based on data from 1,257 censuses and 761 population registry location-years [22].

The present study retrieved estimated numbers of deaths and population sizes by sex, age group, cause of death, and country from 1990 to 2017 from GBD 2017. Based on income categories defined by the World Bank in 2017 [23], we also classified the 195 countries and territories into high-income, upper-middle-income, lower-middle-income, and low-income. We used the level 3 categorization of causes of death from GBD 2017, which includes 169 causes of death [15]. Populations were partitioned into 20 age groups from under 5 years to 95 years and older, with each age group spanning 5 years.

### Decomposition method

Several methods have been developed to decompose differences in the total number of deaths into contributions from 3 components: population growth, population ageing, and mortality change [4,15–17]. Each method has pros and cons, but most are sensitive to the choice of decomposition order and the choice of reference group, yielding inconsistent or even conflicting results from the same data [18]. Recently, we developed a decomposition method that overcomes these limitations by calculating the contributions of the 3 components based on the following formulas [18]:

$$M_{\mathrm{p}} = \sum_{i=1}^{20}(N_2 - N_1)s_{i1}m_{i1} \tag{1}$$

$$M_{\mathrm{a}} = \sum_{i=1}^{20}N_1(s_{i2} - s_{i1})m_{i1} \tag{2}$$

$$M_{\mathrm{m}} = \sum_{i=1}^{20} N_1 s_{i1}(m_{i2} - m_{i1}) \tag{3}$$

$$I_{\mathrm{pa}} = \sum_{i=1}^{20} (N_2 - N_1)(s_{i2} - s_{i1})m_{i1} \tag{4}$$

$$I_{\mathrm{pm}} = \sum_{i=1}^{20} (N_2 - N_1)s_{i1}(m_{i2} - m_{i1}) \tag{5}$$

$$I_{\mathrm{am}} = \sum_{i=1}^{20} N_1(s_{i2} - s_{i1})(m_{i2} - m_{i1}) \tag{6}$$

$$I_{\mathrm{pam}} = \sum_{i=1}^{20} (N_2 - N_1)(s_{i2} - s_{i1})(m_{i2} - m_{i1}) \tag{7}$$

where $M_{\mathrm{a}}$, $M_{\mathrm{p}}$, and $M_{\mathrm{m}}$ indicate the main effects of the 3 components population ageing, population growth, and mortality change, respectively; $I_{\mathrm{pa}}$, $I_{\mathrm{pm}}$, $I_{\mathrm{am}}$, and $I_{\mathrm{pam}}$ denote the 2-way and 3-way interactions of the 3 components; $m_{ij}$ and $s_{ij}$ denote the age-specific mortality rate and proportion of population, respectively, for the $i$th age group in the $j$th year ($i = 1, 2, \ldots, 20$; $j = 1, 2$); and $N_1$ and $N_2$ represent the population size for group 1 and group 2, respectively. The change in the number of deaths can then be attributed to population ageing, population growth, and change of age-specific mortality rate as follows:

$$A = M_{\mathrm{a}} + \tfrac{1}{2}I_{\mathrm{pa}} + \tfrac{1}{2}I_{\mathrm{am}} + \tfrac{1}{3}I_{\mathrm{pam}} \tag{8}$$

$$P = M_{\mathrm{p}} + \tfrac{1}{2}I_{\mathrm{pa}} + \tfrac{1}{2}I_{\mathrm{pm}} + \tfrac{1}{3}I_{\mathrm{pam}} \tag{9}$$

$$M = M_{\mathrm{m}} + \tfrac{1}{2}I_{\mathrm{pm}} + \tfrac{1}{2}I_{\mathrm{am}} + \tfrac{1}{3}I_{\mathrm{pam}} \tag{10}$$

Details about the method are provided in S1 Text. All data analyses were performed using R 3.6.0, and the package "maps" was used to draw the maps.

## Data analysis

Using the decomposition method described above, we calculated the absolute and relative contributions of the 3 components (population growth, population ageing, and mortality change) to the difference in number of total deaths and subgroup deaths between 1990 and each year from 1991 to 2017 for the global population as well as for each country/territory included in this study. The absolute contribution was the number of attributed deaths, while the relative contribution ("attributed proportion") was estimated as the number of attributed deaths divided by the total number of deaths in 1990 × 100%. A positive contribution indicates an increase in total deaths, while a negative contribution indicates a decrease in total deaths.

We plotted the absolute contributions of the 3 components to the changes in total deaths from 1991 to 2017. The relative contributions of population ageing across the study time period were graphed by sex for the world and for the 4 World Bank income categories. We tabulated the 10 causes of death with the greatest increase in the number of deaths associated with population ageing between 1990 and 2017 by sex, as well as the 10 causes of death with the greatest decrease in number of deaths related to population ageing. We estimated country-specific relative contributions of population ageing from 1990 to 2017 by sex and cause of death.

Last, for countries where population ageing was associated with increases in deaths between 1990 and 2017, we calculated the ratio of the number of deaths attributed to mortality change to that attributed to population ageing ($R$) to assess the comparative contributions of mortality

changes (reductions in most countries) versus population ageing to changes in total deaths. *R* < −1 suggests that the effect of mortality decrease in reducing the total deaths is larger than the effect of population ageing in increasing the total deaths. *R* = −1 indicates that the effects of mortality reduction and of population ageing are equal and thus offset each other, and −1 < *R* < 0 suggests the effect of mortality reduction is less than that of population ageing. Finally, *R* > 0 suggests changes in mortality rates and population ageing were related to increases in deaths between 1990 and 2017. All analyses were stratified by sex because the impact of population ageing differs between males and females [4,8].

We finalized the analysis strategies in June 2019, including exploring patterns in deaths attributed to population ageing, variation in number of attributed deaths, and change in number of attributed deaths across sex, country income category, and cause of death, as well as comparing the effect of mortality change to the effect of population ageing. Data analyses were completed June–August 2019. This research was performed and reported adhering to the Guidelines for Accurate and Transparent Health Estimates Reporting (GATHER) statement (S1 GATHER Checklist) [24].

## Results

### Population ageing

According to GBD 2017 population estimates, the number of people aged 65 years and older increased by 105% globally from 1990 (327.6 million) to 2017 (673.7 million); the number of global deaths increased from 19.1 million to 32.2 million in the same time period (Table 1). The proportion of people aged 65 years and older increased from 12.1% (121.5 million) to 17.5% (208.6 million), from 5.6% (119.0 million) to 10.3% (270.5 million), and from 3.9% (75.1 million) to 5.5% (170.6 million) between 1990 and 2017 in high-, upper-middle-, and lower-middle-income countries, respectively, but decreased from 3.2% (10.7 million) to 3.1% (20.8 million) in low-income countries. Accordingly, the proportion of people younger than 30 years was much lower and decreased more in high-, upper-middle-, and lower-middle-income countries (from 44.7% [447.1 million] to 35.1% [4,17.8 million], 59.5% [1,256.9 million] to 41.5% [1,096.3 million], and 66.5% [1,282.4 million] to 58.3% [1,819.3 million], respectively) between 1990 and 2017, compared to that in low-income countries (from 71.9% [236.8 million] to 70.6% [470.6 million]). The changes in the proportion of people aged 65 years and older for each country/territory are shown in S1 Table.

### Change in global deaths attributed to population ageing

Using 1990 as the baseline, the increase in the number of global deaths attributed to population ageing grew gradually from 1991 to 2017 and reached 12 million in 2017 (Fig 1). Between 1990 and 2017, population growth was associated with an increase of 12 million deaths, while mortality change (i.e., reductions in most age-specific mortality rates) was associated with a decrease of 21 million deaths.

The proportion of deaths attributed to population change rose steadily between 1991 and 2017 for the world overall and for high-, upper-middle-, and lower-middle-income countries (Fig 2). The attributed proportion increased more sharply in high-, upper-middle-, and lower-middle-income countries compared to low-income countries. These patterns were similar for both males and females. The attributed proportion among males was 27.9% (7.0 million) for the world and was 51.2% (2.3 million), 56.2% (4.5 million), 12.2% (1.2 million), and −7.4% (−0.2 million) for high-, upper-middle-, lower-middle-, and low-income countries between 1990 and 2017, respectively (Fig 2A). The corresponding numbers globally and by country income category for females were 26.0% (5.6 million), 50.6% (2.1 million), 55.6% (3.7 million),

**Table 1.  Number of population (in millions) and deaths (in millions) in 1990 and 2017 globally and by country income category.**

| Variable | Value | Global | | Country income category | | | | | | | |
|---|---|---|---|---|---|---|---|---|---|---|---|
| | | | | High | | Upper-middle | | Lower-middle | | Low | |
| | | 1990 | 2017 | 1990 | 2017 | 1990 | 2017 | 1990 | 2017 | 1990 | 2017 |
| **Population** | | | | | | | | | | | |
| Sex | | | | | | | | | | | |
| Male | Number | 2,717.5 | 3,834.5 | 491.8 | 591.5 | 1,067.3 | 1,322.2 | 983.7 | 1,576.3 | 162.5 | 330.1 |
| | Percent | 50.4% | 50.2% | 49.2% | 49.7% | 50.6% | 50.2% | 51.0% | 50.5% | 49.3% | 49.5% |
| Female | Number | 2,677.2 | 3,806.0 | 508.8 | 598.1 | 1,044.1 | 1,312.7 | 945.6 | 1,544.5 | 167.1 | 336.7 |
| | Percent | 49.6% | 49.8% | 50.8% | 50.3% | 49.4% | 49.8% | 49.0% | 49.5% | 50.7% | 50.5% |
| Age group | | | | | | | | | | | |
| <30 years old | Number | 3,237.4 | 3,815.1 | 447.1 | 417.8 | 1,256.9 | 1,096.3 | 1,282.4 | 1,819.3 | 236.8 | 470.6 |
| | Percent | 60.0% | 49.9% | 44.7% | 35.1% | 59.5% | 41.6% | 66.5% | 58.3% | 71.9% | 70.6% |
| 30–64 years old | Number | 1,829.8 | 3,151.7 | 431.9 | 563.1 | 735.5 | 1,268.1 | 571.8 | 1,130.9 | 82.1 | 175.4 |
| | Percent | 33.9% | 41.3% | 43.2% | 47.3% | 34.8% | 48.1% | 29.6% | 36.2% | 24.9% | 26.3% |
| ≥65 years old | Number | 327.6 | 673.7 | 121.5 | 208.6 | 119.0 | 270.5 | 75.1 | 170.6 | 10.7 | 20.8 |
| | Percent | 6.1% | 8.8% | 12.1% | 17.5% | 5.6% | 10.3% | 3.9% | 5.5% | 3.2% | 3.1% |
| **Number of deaths** | | | | | | | | | | | |
| Sex | | | | | | | | | | | |
| Male | Number | 24.9 | 30.4 | 4.5 | 5.3 | 8.0 | 10.7 | 9.8 | 11.6 | 2.6 | 2.6 |
| | Percent | 53.6% | 54.3% | 51.9% | 50.9% | 54.4% | 56.4% | 53.7% | 54.2% | 53.7% | 54.1% |
| Female | Number | 21.6 | 25.6 | 4.1 | 5.2 | 6.7 | 8.3 | 8.5 | 9.8 | 2.2 | 2.2 |
| | Percent | 46.4% | 45.7% | 48.1% | 49.1% | 45.6% | 43.6% | 46.3% | 45.8% | 46.3% | 45.9% |
| Age group | | | | | | | | | | | |
| <30 years old | Number | 15.6 | 8.2 | 0.4 | 0.2 | 3.5 | 1.1 | 8.6 | 4.6 | 3.0 | 2.2 |
| | Percent | 33.5% | 14.6% | 4.8% | 1.9% | 24.0% | 5.8% | 47.1% | 21.6% | 62.2% | 46.1% |
| 30–64 years old | Number | 11.8 | 15.5 | 1.9 | 1.9 | 4.3 | 5.3 | 4.6 | 7.0 | 1.0 | 1.3 |
| | Percent | 25.4% | 27.8% | 22.4% | 17.7% | 29.2% | 28.0% | 24.9% | 32.5% | 20.8% | 27.3% |
| ≥65 years old | Number | 19.1 | 32.2 | 6.3 | 8.4 | 6.9 | 12.5 | 5.1 | 9.8 | 0.8 | 1.3 |
| | Percent | 41.1% | 57.6% | 72.8% | 80.4% | 46.8% | 66.1% | 27.9% | 45.8% | 17.0% | 26.5% |

14.0% (1.2 million), and −3.2% (−72,000), respectively (Fig 2B). The negative attributed proportion in low-income countries results from decreases in the proportion of people aged 65 years and older (from 3.2% to 3.1%) and the fact that older adults have much higher overall mortality rates than young people.

The impact of population ageing significantly differed across causes of death (Table 2). Among males, across the 169 causes of death, population ageing was most significantly associated with increases in deaths from ischemic heart disease (1.74 million), stroke (1.13 million), chronic obstructive pulmonary disease (0.77 million), and tracheal, bronchial, and lung cancer (0.38 million) between 1990 and 2017. As an accompanying effect of population ageing, the percentage of children in the population decreased gradually, with associated reductions of deaths from neonatal disorders (0.39 million) and congenital birth defects (0.10 million) between 1990 and 2017. The cause-specific decomposition results for females were roughly similar to those for males, although the magnitudes of the contributions were somewhat lower and the ranks slightly differ.

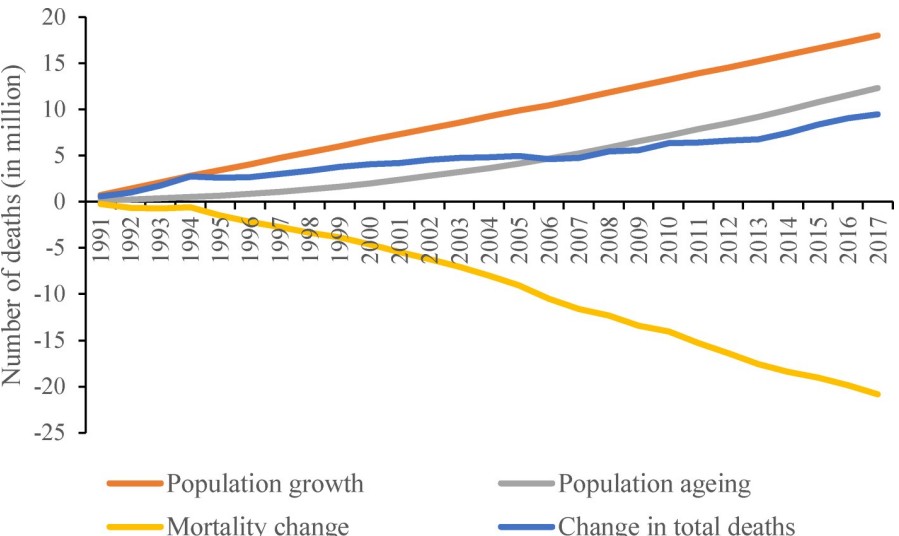

**Fig 1. Global death changes associated with population ageing, population growth, and mortality change from 1990 to 2017.** The decomposition was conducted using the number of deaths in 1990 as the reference for each year.

## Change in country-specific deaths attributed to population ageing

Population ageing was associated with increases in deaths for males in 152 countries and territories between 1990 and 2017, but decreases in deaths for males in 43 countries and territories, primarily in Africa (Fig 3A). The latter results were the consequence of the decreased proportion of people aged 65 years and older and the fact that older age groups have higher all-cause mortality rates than younger age groups. The proportion of changes in male deaths associated with population ageing between 1990 and 2017 ranged from −44% in Afghanistan to 117% in Japan.

The geographic variation in attributed proportion for females differed moderately from that for males (Fig 3B). Among females, population ageing was associated with increases in deaths in 159 countries and territories, and decreases in deaths in 36 countries and territories, between 1990 and 2017. Similar to the pattern for males, Japan had the highest attributed proportion of female deaths (154%) and Afghanistan had the lowest (−30%).

Table 3 shows that ischemic heart disease and stroke were the 2 causes of death that were most adversely affected by population ageing between 1990 and 2017 for both males and females. Fourteen countries had ≥20% net increase in ischemic heart disease deaths attributed to population ageing among males, and 23 countries among females. Most were high-income countries. Two countries/territories had attributed proportions greater than 20% for stroke-related deaths among males (Albania, 22%; South Korea, 22%), and 7 among females (South Korea, 29%; Japan, 26%; Macedonia, 24%; Portugal, 22%; Bosnia and Herzegovina, 22%; Montenegro, 22%; and Taiwan of China, 21%) (S2 Table).

With a few exceptions, the proportion of death increase associated with population ageing between 1990 and 2017 was less than 10% for most diseases in both sexes (Table 3). For males, the attributed proportion was 13% for chronic obstructive pulmonary disease in China, 10% for Alzheimer disease and other types of dementia in Japan, and 12% for lower respiratory infections in Japan. The attributed proportion was greater than 10% for Alzheimer disease and other types of dementia in 13 countries and territories for females. The attributed proportion for chronic obstructive pulmonary disease was 20% in Korea and 14% in China for females. The attributed proportion for diabetes mellitus exceeded 10% in Fiji (14%), Northern Mariana

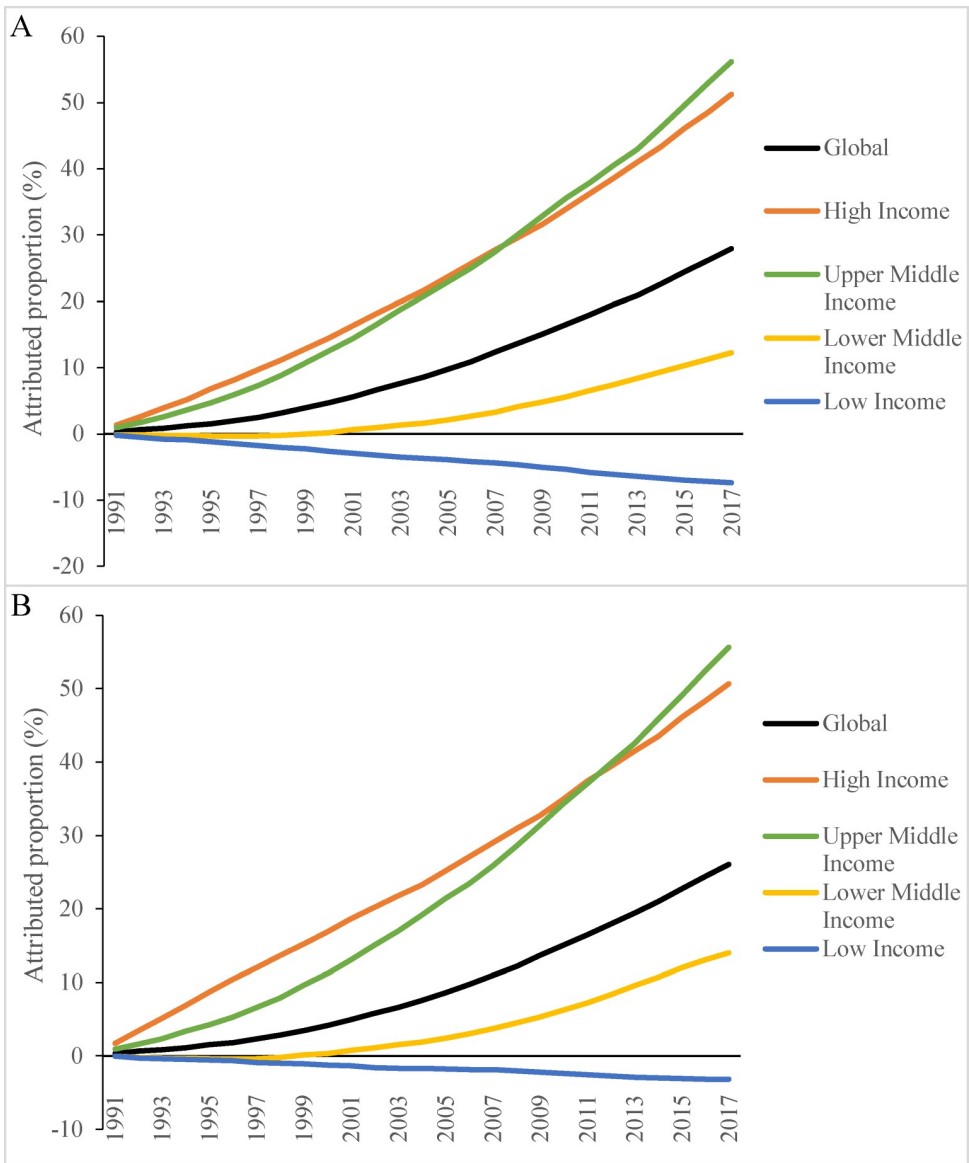

**Fig 2. Proportion of deaths associated with population ageing globally and by country income category, 1990–2017.** (A) Male; (B) female. Decomposition analysis was conducted using the number of deaths in 1990 as the reference. The attributed proportion of deaths was calculated as the number of deaths attributed to population ageing divided by total deaths in 1990 × 100%.

Islands (11%), and Trinidad and Tobago (11%) for females. For chronic kidney disease, only the Northern Mariana Islands (11%) had an attributed proportion greater than 10% for females. Three countries had an attributed proportion exceeding 10% for lower respiratory infections for females: Japan (12%), Singapore (11%), and Andorra (11%) (S2 Table).

## Comparative contributions of mortality reduction versus population ageing

Globally, the decrease in deaths attributed to mortality reduction far exceeded the increase in deaths related to population ageing between 1990 and 2017 (−21 million versus 12 million)

**Table 2. Top 10 causes of death with the highest increase and decrease in the number (in thousands) and proportion associated with population ageing between 1990 and 2017.**

| Rank | Male | | Female | |
|---|---|---|---|---|
| | Cause of death | Number (%) | Cause of death | Number (%) |
| 1 | Ischemic heart disease | 1,735 (7.0%) | Ischemic heart disease | 1,470 (6.8%) |
| 2 | Stroke | 1,126 (4.5%) | Stroke | 1,067 (4.9%) |
| 3 | Chronic obstructive pulmonary disease | 771 (3.1%) | Alzheimer disease and other types of dementia | 621 (2.9%) |
| 4 | Tracheal, bronchial, and lung cancer | 379 (1.5%) | Chronic obstructive pulmonary disease | 516 (2.4%) |
| 5 | Alzheimer disease and other types of dementia | 356 (1.4%) | Hypertensive heart disease | 172 (0.8%) |
| 6 | Tuberculosis | 227 (0.9%) | Diabetes mellitus | 170 (0.8%) |
| 7 | Cirrhosis and other chronic liver diseases | 214 (0.9%) | Breast cancer | 146 (0.7%) |
| 8 | Stomach cancer | 199 (0.8%) | Chronic kidney disease | 137 (0.6%) |
| 9 | Diabetes mellitus | 170 (0.7%) | Tracheal, bronchial, and lung cancer | 135 (0.6%) |
| 10 | Chronic kidney disease | 169 (0.7%) | Colon and rectum cancer | 120 (0.6%) |
| 160 | Typhoid and paratyphoid | −15 (−0.1%) | Typhoid and paratyphoid | −14 (−0.1%) |
| 161 | Whooping cough | −17 (−0.1%) | Drowning | −14 (−0.1%) |
| 162 | Drowning | −20 (−0.1%) | Tetanus | −14 (−0.1%) |
| 163 | Sexually transmitted infections excluding HIV | −21 (−0.1%) | Whooping cough | −21 (−0.1%) |
| 164 | Protein-energy malnutrition | −22 (−0.1%) | Meningitis | −26 (−0.1%) |
| 165 | Meningitis | −28 (−0.1%) | Protein-energy malnutrition | −27 (−0.1%) |
| 166 | Malaria | −53 (−0.2%) | Malaria | −51 (−0.2%) |
| 167 | Measles | −55 (−0.2%) | Measles | −57 (−0.3%) |
| 168 | Congenital birth defects | −100 (−0.4%) | Congenital birth defects | −91 (−0.4%) |
| 169 | Neonatal disorders | −390 (−1.6%) | Neonatal disorders | −314 (−1.5%) |

The attributed proportion for males was calculated as the number of deaths attributed to population ageing for each cause of death/24.9 million (total male deaths in 1990) × 100%. The attributed proportion for females was calculated as the number of deaths attributed to population ageing for each cause of death/21.6 million (total female deaths in 1990) × 100%.

(Fig 1). In fact, the ratio ($R$) of the decrease in deaths (a negative change) attributed to mortality reduction to the increase in deaths (a positive change) related to population ageing between 1990 and 2017 was −1.6 for males and −1.8 for females. The ratio differed greatly across sex and country income categories; it was −1.0, −0.8, and −3.8 for males in high-, upper-middle-, and lower-middle-income countries, respectively, and −0.9, −1.0, and −3.7 for females in high-, upper-middle-, and lower-middle-income countries, respectively. Because the proportion of people aged 65 years and older decreased in most low-income countries, we did not analyze the comparative contributions of mortality reduction versus population ageing for this category.

Of the 152 countries that experienced an increase in male deaths related to population ageing between 1990 and 2017, 77 countries and territories (51%) had $R \leq −1$, 66 (43%) had $−1 < R \leq 0$, and 9 (6%) had $R > 0$ (Guam, Jamaica, Lesotho, North Korea, Swaziland, Syria, Ukraine, US Virgin Islands, and Uzbekistan) (Fig 4A). The lowest ratio was in Eritrea (−161), and the highest was in Lesotho (16).

Among females, 159 (82%) countries and territories experienced an increase in deaths attributed to population ageing between 1990 and 2017 (Fig 4B). Of these, 76 (48%) had $R \leq −1$, 78 (49%) had $−1 < R \leq 0$, and 5 (3%) had $R > 0$ (American Samoa, Guam, Lesotho, Serbia, and Swaziland). The lowest ratio was in Zambia (−119), and the highest was in Lesotho (4). Detailed country-specific ratios appear in S3 Table.

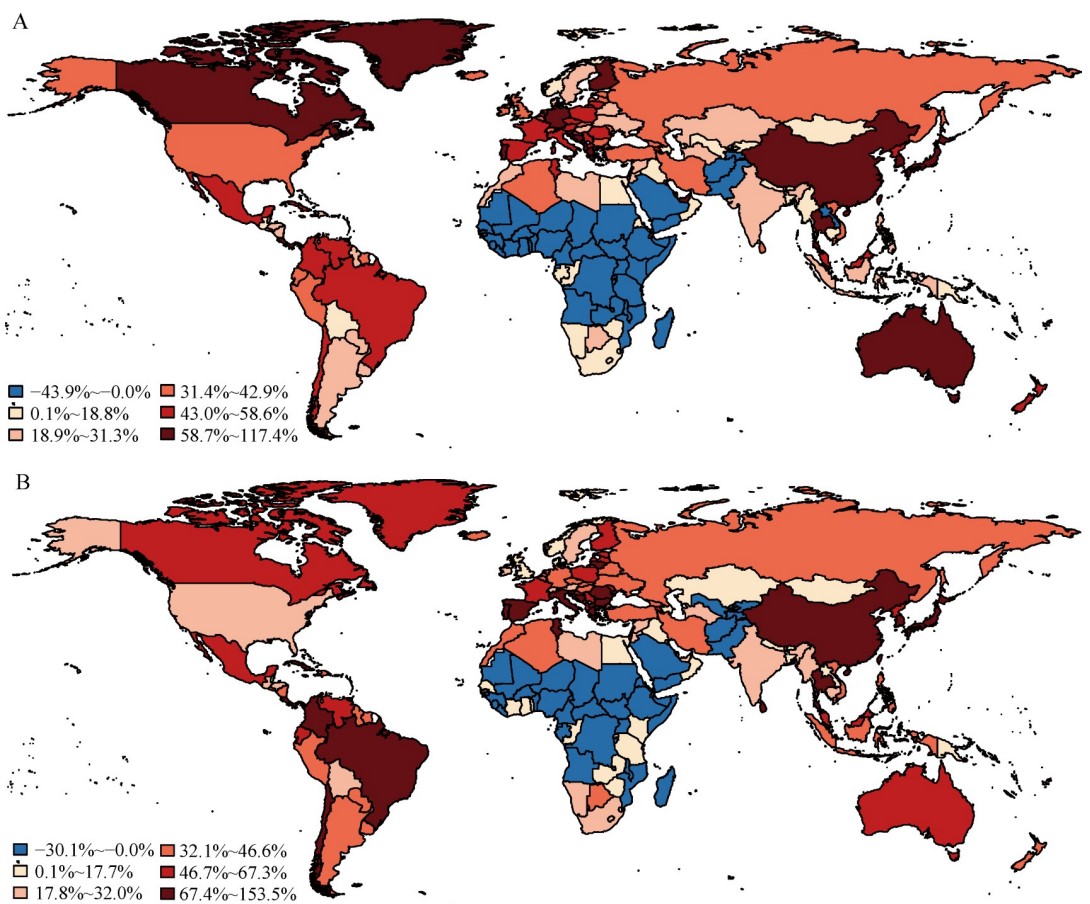

**Fig 3. Proportion of deaths associated with population ageing between 1990 and 2017 in 195 countries and territories.** (A) Male; (B) female. The attributed proportion was calculated as the change in total deaths attributed to population ageing between 1990 and 2017 divided by total deaths in 1990 × 100%. Countries and territories with negative attributed proportions were treated as a single category. Countries with positive attributed proportions were classified into 5 categories according to quintiles of positive attributed proportions. The maps were drawn using the R package "maps," which was based on the data from the Natural Earth project.

## Discussion

In this study, we reported on global death changes from 1990 to 2017 attributed to population ageing for 169 causes of death both globally and by country/territory using a decomposition method. We have 4 key findings. First, population ageing was associated with an increase of 12 million deaths worldwide between 1990 and 2017. The death increases occurred primarily in high-, upper-middle-, and lower-middle-income countries; in fact, many low-income countries experienced decreases in deaths attributed to population ageing. Second, between 1990 and 2017, most of the increases in deaths related to population ageing were from ischemic heart disease (1.74 million for males and 1.47 million for females) and stroke (1.13 million for males and 1.07 million for females). Third, the impact of population ageing greatly varied across countries and territories, causing increases in deaths in most countries but decreases in deaths in some countries. The country-specific impacts also differed by cause of death. Last, the increase in deaths related to population ageing between 1990 and 2017 was outweighed by the decrease in deaths attributed to mortality reduction both globally (−21 million versus 12

**Table 3. Number of countries and territories with different increases in cause-specific proportions of deaths associated with population ageing between 1990 and 2017.**

| Sex and cause of death | Increase in attributed proportion of deaths (number of countries/territories) | | | | |
|---|---|---|---|---|---|
| | 1%–4% | 5%–9% | 10%–14% | 15%–19% | ≥20% |
| **Male** | | | | | |
| Ischemic heart disease | 29 | 43 | 38 | 29 | 14 |
| Stroke | 79 | 51 | 10 | 4 | 2 |
| Chronic obstructive pulmonary disease | 110 | 8 | 1 | 0 | 0 |
| Alzheimer disease and other types of dementia | 101 | 4 | 1 | 0 | 0 |
| Tracheal, bronchial, and lung cancer | 87 | 13 | 0 | 0 | 0 |
| Chronic kidney disease | 75 | 2 | 0 | 0 | 0 |
| Cirrhosis and other chronic liver diseases | 76 | 1 | 0 | 0 | 0 |
| Diabetes mellitus | 68 | 8 | 0 | 0 | 0 |
| Lower respiratory infections | 69 | 6 | 1 | 0 | 0 |
| Prostate cancer | 69 | 3 | 0 | 0 | 0 |
| Colon and rectum cancer | 59 | 0 | 0 | 0 | 0 |
| Stomach cancer | 55 | 2 | 0 | 0 | 0 |
| Hypertensive heart disease | 43 | 0 | 0 | 0 | 0 |
| Road injuries | 26 | 1 | 0 | 0 | 0 |
| Tuberculosis | 27 | 0 | 0 | 0 | 0 |
| **Female** | | | | | |
| Ischemic heart disease | 40 | 40 | 29 | 25 | 23 |
| Stroke | 61 | 57 | 23 | 8 | 7 |
| Alzheimer disease and other types of dementia | 76 | 41 | 10 | 1 | 2 |
| Chronic obstructive pulmonary disease | 92 | 5 | 1 | 0 | 1 |
| Diabetes mellitus | 82 | 13 | 3 | 0 | 0 |
| Breast cancer | 87 | 2 | 0 | 0 | 0 |
| Chronic kidney disease | 77 | 2 | 1 | 0 | 0 |
| Hypertensive heart disease | 79 | 1 | 0 | 0 | 0 |
| Lower respiratory infections | 67 | 6 | 3 | 0 | 0 |
| Colon and rectum cancer | 57 | 0 | 0 | 0 | 0 |
| Cirrhosis and other chronic liver diseases | 45 | 0 | 0 | 0 | 0 |
| Tracheal, bronchial, and lung cancer | 38 | 2 | 0 | 0 | 0 |
| Stomach cancer | 38 | 0 | 0 | 0 | 0 |
| Cervical cancer | 37 | 0 | 0 | 0 | 0 |
| Other cardiovascular and circulatory diseases | 23 | 0 | 0 | 0 | 0 |
| Atrial fibrillation and flutter | 22 | 0 | 0 | 0 | 0 |
| Cardiomyopathy and myocarditis | 19 | 1 | 0 | 0 | 0 |

The attributed proportion was calculated as the number of deaths attributed to population ageing for each cause of death between 1990 and 2017 divided by total deaths in 1990 × 100% for males and females, respectively. Diseases with an attributed proportion of 0% to <1% and diseases with an attributed proportion of ≥1% in less than 20 countries and territories are omitted.

million) and in about half of the countries and territories that experienced an increase in deaths attributed to population ageing.

This study offers a comprehensive set of estimates concerning the health impact of population ageing. Previous publications reported that population ageing was associated with increases in deaths from ischemic heart disease, chronic kidney disease, and cardiovascular deaths globally [17,25] and with deaths from coronary heart disease and musculoskeletal disorders in selected countries [11,26]. Our analyses are distinct in 2 primary ways: (a) examining

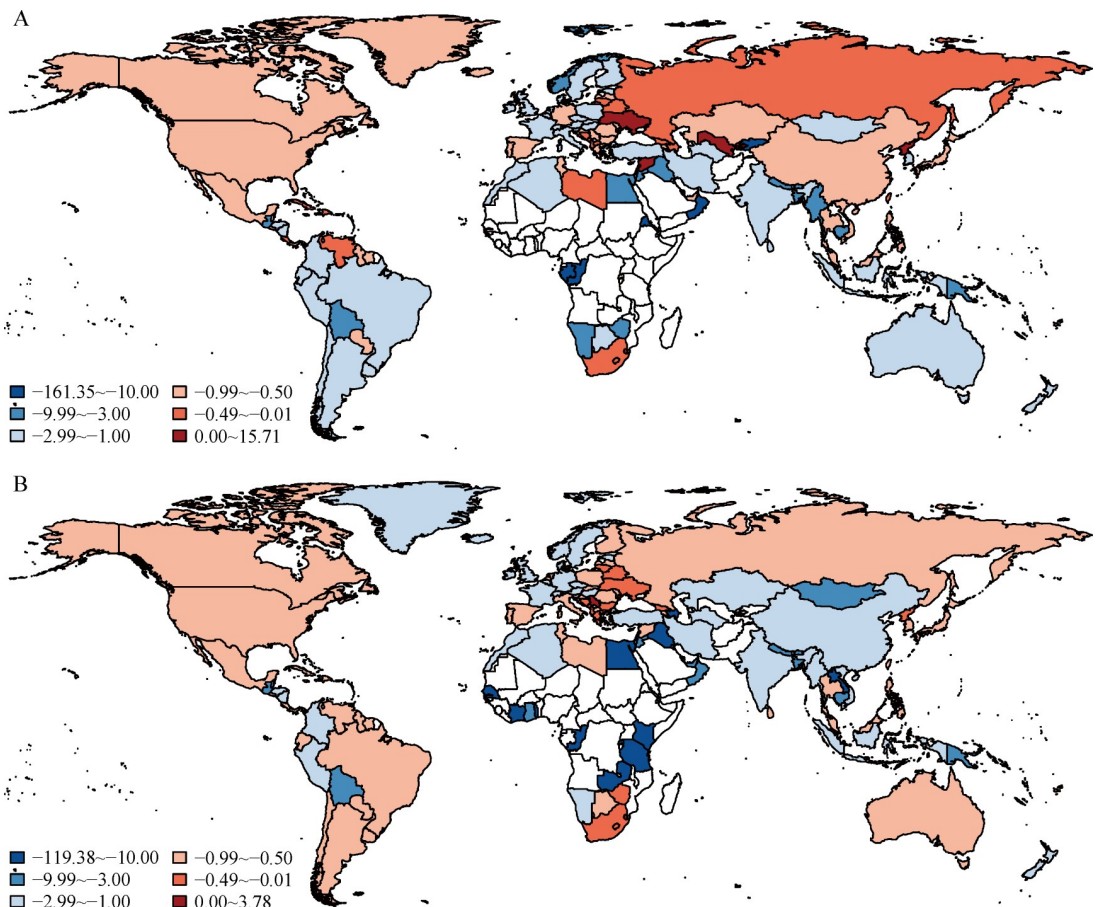

**Fig 4. Ratio between total deaths attributed to change in mortality rate and total deaths associated with population ageing between 1990 and 2017.** (A) Male; (B) female. The ratio was calculated as the change in total deaths attributed to change in mortality rate divided by that associated with population ageing. Blue signifies countries and territories for which the decrease in total deaths attributed to changes in mortality was more than the increase attributed to population ageing between 1990 and 2017. Red signifies countries and territories for which the decrease in total deaths attributed to changes in mortality was less than the increase associated with population ageing between 1990 and 2017. White signifies countries and territories not experiencing an increase in deaths associated with population ageing between 1990 and 2017. The maps were drawn using the R package "maps," which was based on the data from the Natural Earth project.

global deaths attributed to population ageing for 169 causes of death across 195 countries and territories and (b) using a decomposition method that is independent of the choice of decomposition order and reference group, to generate more robust estimates. Consistent with previous findings [11,17,25,26], this study demonstrates heterogeneous health impacts of population ageing across countries. We report increased death burden in many countries but reduced death burden in some countries (typically low-income countries/territories). The contrasting results are likely due to different changes in the age structure of populations across countries/territories (see Tables 1 and S1), as well as great variations in mortality rates across age groups.

Among our notable findings is the fact that population ageing/de-ageing was associated with decreases in deaths from some diseases (for example, ischemic heart disease) in some low-income countries, such as Afghanistan, Liberia, and Guinea, despite being associated with increases in deaths from these diseases globally [17]. Such results likely reflect differences in demographic changes across nations. The proportion of people aged 65 years and older

increased from 12.1% to 17.5% in high-income countries but decreased from 3.2% to 3.1% in low-income countries between 1990 and 2017 [20]. Thus, population ageing/de-ageing was associated with varying health impacts across countries and territories. International organizations and national governments should weigh these variations when developing and implementing action plans to improve health, or tailoring prevention programs to face the potential health impact from population ageing in particular regions and countries.

A key strength of this study was our use of a decomposition method that functions independently of the choice of decomposition order and reference group to comprehensively estimate the health impact of population ageing from 1990 to 2017 for the whole world as well as for 195 individual countries/territories. We considered both all-cause mortality and cause-specific mortality for 169 causes of death. Our decomposition results can be compared across countries/territories and across causes of death, regardless of decomposition order of the factors and the choice of reference group. In addition, we evaluated the extent to which changes in mortality rate alleviated or exceeded the increases in deaths related to population ageing, exploring the importance of prevention efforts to reduce age-specific mortality.

This study has several limitations. First, our results depend on the quality of the estimates for the numbers of deaths and population sizes from GBD 2017. They are therefore affected by factors that impact the accuracy of the GBD 2017 estimates, including lack of high-quality mortality and migration data for some countries and lack of widely validated estimation methods [15,22]. Second, we cannot provide 95% confidence intervals for our estimates because we are unable to access the full posterior samples of cause-specific mortality rates stratified by age, sex, location, and year from the GBD 2017 study [5,15,22]. Third, population ageing can be caused both by decreasing fertility rates and by increasing life expectancy [2,27]. The method used in this study does not explore the 2 mechanisms of population ageing. Fourth, the method used in this study only considers 3 factors, and thus ignores any heterogeneity underlying other factors related to changes in total mortality. For example, temporal changes in age structure or mortality rates may vary by sex or income level. We conducted analyses specific to each subpopulation defined by sex, country income level, and cause of death and thus partially accounted for such heterogeneity. These limitations could be overcome through methodological innovations and improving data quality in future research.

This study offers valuable data and insights to guide health policy-making and reform of health systems, especially in countries experiencing rapid population ageing such as South Korea, Japan, and China. Our results demonstrate the success and promise of disease prevention and health promotion efforts. Encouragingly, the increase in deaths related to population ageing was outweighed by the decrease in deaths attributed to mortality rate reductions between 1990 and 2017. This was true both globally and in about half the countries and territories studied. The challenges brought about to society through population ageing can therefore be substantially alleviated through disease prevention and health promotion. Despite an ageing global population, mortality rates worldwide are decreasing [8]. To maintain these successes, health resources should be allocated to further reduce mortality rates in countries/territories where the effect of population ageing much outweighed that of mortality reduction, as illuminated by our findings. As lower income countries experience economic, infrastructure, and public health improvements, they may face challenges from population ageing similar to the challenges higher income nations are now confronting. They should benefit from the lessons learned in higher income countries, and should invest in proven interventions to promote healthy ageing [28–30]. As an example, scholars have highlighted the successful efforts of Canada in promoting active, healthy ageing through strategies such as collaborating with various stakeholders to advocate physical activity, and have argued that these efforts could readily be adapted and disseminated in sub-Saharan African countries [28]. Our results help identify

countries with successful experiences, especially those with reduced age-specific mortality outweighing population ageing, in shaping the long-term pattern of death burden.

## Conclusions

This study identified a global pattern of increased disease-related deaths attributed to population ageing from 1990 to 2017. Because of heterogeneity in age structure and age-specific mortality rates, the impact of population ageing on deaths varied by sex, country income level, country, and cause of death. The increase in deaths related to population ageing was largely offset by mortality reductions both globally and in about half of individual countries and territories. To respond to the increase in deaths related to population ageing for some causes of death, policy-makers should invest more in preventive medicine, ageing-related health research, and implementation of proven cost-effective interventions against chronic diseases and injuries.

## Supporting information

**S1 GATHER Checklist. Guidelines for Accurate and Transparent Health Estimates Reporting checklist.**
(DOCX)

**S1 Table. Proportion of people aged 65 years and older in 1990 and 2017.**
(DOCX)

**S2 Table. Causes of death with proportion of deaths associated with population ageing more than 10%.**
(DOCX)

**S3 Table. Comparative contributions of mortality reduction versus population ageing to change in number of deaths between 1990 and 2017.**
(DOCX)

**S1 Text. The decomposition method.**
(DOCX)

## Author Contributions

**Conceptualization:** Xunjie Cheng, Guoqing Hu.

**Formal analysis:** Xunjie Cheng.

**Methodology:** Xunjie Cheng, Yang Yang, Guoqing Hu.

**Software:** Xunjie Cheng.

**Validation:** David C. Schwebel, Zuyun Liu, Li Li, Peixia Cheng, Peishan Ning.

**Writing – original draft:** Xunjie Cheng.

**Writing – review & editing:** Yang Yang, David C. Schwebel, Zuyun Liu, Li Li, Peixia Cheng, Peishan Ning, Guoqing Hu.

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
