## [Decision Letter · Decision Letter 0]

29 Jan 2020

Dear Dr. Hu,

Thank you very much for submitting your manuscript "Impact of population ageing on global deaths, 1990-2017" (PMEDICINE-D-19-03747) for consideration at PLOS Medicine. 

Your paper was evaluated by a senior editor and discussed among the editors here. It was also discussed with an academic editor with relevant expertise, and sent to independent reviewers, including a statistical reviewer. The reviews are appended at the bottom of this email and any accompanying reviewer attachments can be seen via the link below:

[LINK]

In light of these reviews, I am afraid that we will not be able to accept the manuscript for publication in the journal in its current form, but we would like to consider a revised version that addresses the reviewers' and editors' comments. Obviously we cannot make any decision about publication until we have seen the revised manuscript and your response, and we plan to seek re-review by one or more of the reviewers. 

We expect to receive your revised manuscript by Feb 12 2020 11:59PM. Please email us (plosmedicine@plos.org) if you have any questions or concerns.

We look forward to receiving your revised manuscript. 

Sincerely,

Louise Gaynor-Brook, MBBS PhD

Associate Editor

on behalf of:

Thomas McBride, PhD

Senior Editor 

PLOS Medicine

plosmedicine.org

General comments: The flow of the main text is, at times, difficult to follow. It would be appreciated if your manuscript could be proofread thoroughly by someone with full professional proficiency in English prior to resubmission.

Please remove indents from the beginning of paragraphs.

Please revise your title according to PLOS Medicine's style, placing the study design in the subtitle (ie, after a colon). We suggest "Population ageing and mortality during 1990-2017: a global cross-sectional analysis" or similar.

Abstract Background: Please expand upon the context of why the study is important. The final sentence should clearly state the study question.

Please combine the Methods and Findings components of your Abstract under ‘Methods and Findings’

Please include the study design, number of individuals included in the GBD 2017 dataset, and brief demographic details e.g. age, sex, distribution of data according to income categories, etc. 

Please expand upon the results relating to variation due to economic development levels and geographic regions

Line 38 - Please report your results for increases and decreases in deaths in separate sentences. 

Please include the important dependent variables that are adjusted for in the analyses.

In the last sentence of the Abstract Methods and Findings section, please describe the main limitation(s) of the study's methodology.

Please begin your Abstract Conclusions with “"In this study, we observed ..." or similar. Please address the study implications substantiated by the results, emphasizing what is new without overstating your conclusions.

At this stage, we ask that you include a short, non-technical Author Summary of your research to make findings accessible to a wide audience that includes both scientists and non-scientists. The Author Summary should immediately follow the Abstract in your revised manuscript. This text is subject to editorial change and should use non-identical language distinct from the scientific abstract. Please see our author guidelines for more information: https://journals.plos.org/plosmedicine/s/revising-your-manuscript#loc-author-summary

Introduction

Please address past research and explain the need for and potential importance of your study. Indicate whether your study is novel and how you determined that. If there has been a systematic review of the evidence related to your study (or you have conducted one), please refer to and reference that review and indicate whether it supports the need for your study. 

Line 65 - please expand upon the importance of an increase in global disability adjusted life years (DALYs) 

Line 88 - please revise ‘currently lacking’ and ‘novel’ to avoid assertions of primacy 

Methods 

Please report your data according to GATHER and enclose a completed GATHER checklist as a supplementary document. See http://gather-statement.org/ In the checklist please include sufficient text excerpted from the manuscript to explain how you accomplished all applicable items. When completing the checklist, please use section titles and paragraph numbers, rather than page numbers. 

Did your study have a prospective protocol or analysis plan? Please state this (either way) early in the Methods section. If a prospective analysis plan was used in designing the study, please include the relevant prospectively written document with your revised manuscript as a Supporting Information file to be published alongside your study, and cite it in the Methods section. If no such document exists, please make sure that the Methods section transparently describes when analyses were planned, and when/why any data-driven changes to analyses took place. 

Results

Please include as Table 1 a summary of the number of individuals included in the GBD 2017 dataset, and baseline demographic data e.g. age, sex, distribution of data according to income categories, etc. 

Line 203 - please revise to ‘bronchial’

Line 219 - please expand upon how changes in male deaths “can be explained by increases in the percentage of younger population” 

Line 242 - please revise sentence beginning ‘Fourteen (23) countries’ to better distinguish between results presented for males and females

Lines 253, 254 - please revise to ‘other types of dementia’

Lines 255 - 261: please clarify which sex these results apply to

Figure 3 - Please explain why range of ‘negative proportions’ (shown in blue) does not extend to zero, and why range of first quintile of the positive proportions begins at 0.7% / 0.8% (and not 0.1) 

Figure 4 - please revise title to “Ratios between total deaths attributed to mortality change and deaths attributed to population ageing…”

Please present numerators and denominators for percentages in the Tables.

Table 1 - please revise to ‘bronchial’

Table 2 - please revise to ‘other types of dementia’ ; ‘bronchial’ ; and ‘pulmonary’

Table S2 - please revise to ‘other types of dementia’

Discussion 

Please remove subheadings from within your Discussion.

Please begin your Discussion with "In this study” or similar

Please present and organize the Discussion as follows: a short, clear summary of the article's findings; what the study adds to existing research and where and why the results may differ from previous research; strengths and limitations of the study; implications and next steps for research, clinical practice, and/or public policy; one-paragraph conclusion.

Line 315 - please revise to ‘have caused’

Line 352 - please revise ‘high-risk age groups differ across diseases’

Comments from the reviewers:

Reviewer #1: I mostly confine my remarks to statistical aspects of this paper.

However, as someone who is new to this idea of disagregating these effects, I was a little unclear as to the purpose of the whole exercise. I guess other people already know this, but, since the audience for PLoS will surely include other people like me, it would be good to give a bit more about this in the introduction. I can certainly see why we would want to know, e.g., projections of disease rates and counts in various countries. But I'm not so clear as to the need for knowing what proportion of the number of deaths is due to aging vs. other causes.

Now, to statistics. These were generally fine. I would like to have a bit more detail in the appendix about the derivation of the formulas. I spent a good bit of time figuring out what was being done and I think it is correct, but the authors could help by spelling out how the various formulas were derived and why they are the way they are. 

Line 45-47: I'm not sure how this follows from the results. I don't disagree with the conclusion, but couldn't we have concluded this a priori? What possible results would make this conclusion incorrect?

Line 101-102 Some detail of what was done would be good.

Line 120-124: It won't really affect anything, but I think the subscripts and formula labels could be better chosen, just to make it easier to follow. There are three factors: Aging, population growth and mortality change. Why not use A, P and M for these, both instead of A, B and C and instead of S, P and M (in the subscripts)?

Fig 1 Stacked bar charts (which is really what this is) aren't a great method. See the work of William S. Cleveland. I would use a line graph with year on the x axis, deaths on the y axis and three or four lines (one for each cause and maybe one for total)

Fig 2: First, the y axis can't be proportion of deaths - proportions have to add to 1.00 (or, percentages to 100). I'm not sure what is being graphed here. Is it number of deaths? Second, I'd put the labels for the lines next to the lines on the right axis for easier reading.

Peter Flom

Reviewer #2: This work tries to provide a better understanding of the drivers causing (old age) mortality (line 22-23). Conditional on the choice of their models, there is no reason to dispute the data presented. Questions arise however on the framing of work, the underlying assumptions, and the interpretation of the data.

Framing

This work is positioned in a reasoning that 'ageing is a global public health challenge' (line 21-22). The direct connect between population ageing and public health challenge however, is a postulation, not a fact. The idea that aging is a (negative) 'challenge' is adhered to by many, but the question is whether it is correct, and or helpful. The key is that a decrease in the force of mortality is a good thing: it is getting better for individuals. It indicates societal progress and achievements, and allows for longer investments into work, society, others. Rephrased otherwise, population ageing can equally be framed as something positive, e.g. the 'silver economy'. A negative preconception could well be considered a form of ageism. 

Assumptions

Not separating the average age of the population and age specific force of mortality can make up every outcome one can think of. It is the explanation why different methods come up with inconsistent results (line 112-117). It is for these reasons that the authors take an absolute stance, 'attributing differences or changes in total deaths to the changes of various components, or factors, such as population size, age structure and mortality rates' (line 117 onwards). The methodology followed is correct when the absolute number of deaths is the primary endpoint to be considered, e.g. Figure 1. However, it is questionable whether the decomposition method provides 'the robust model' that is claimed to unravel the drivers of mortality. It can easily be deduced from the figure that the change in total deaths is the net result of population size, force of mortality and age structure, phenomena that are interrelated, but very different between countries depending on what happened over successive generations, calendar time and lifespan. The model allows for including these various interactions, but the consequence is that outcomes of the model are only weighted averages, and different for the whole world, regions, and each country separately. When very strict, this underlying heterogeneity between countries and regions would prohibit statistical pooling…

A similar reasoning can be set up when classifying a nation as an "ageing country", even when it is according to the United Nations standard (proportion of people ≥ 65 years old exceeds 7%). Given the huge differences in age specific mortality, morbidity, functionality (e.g. pension age varies between 55 to none) 7% can be appreciated as low, preferred, or high. Moreover, the use of 'ageing countries' is ambivalent as there may be different underlying processes going on. In extreme, one could argue that some nations that have successfully dealt with early mortality at young age are now finally ageing, which is a positive thing. This is not a semantic discussion, e.g. a careful exploration and presentation of these separate phenomena unmasks the double burden of disease, and explains why these nations can only prosper economically when their populations are ageing.

Interpretation

It is very difficult to infer 'global trend in disease-related deaths attributed to population ageing from 1990 to 2017' (line 399). The reason is that the underlying demographic dynamics are so different:

- First, in some countries it could be that: As the number of new-borns decreases, population ageing accelerates as a global public health challenge. The interpretation being that it is the decrease in fertility that is the underlying problem and should be addressed, which reasoning, correct or wrong, is now being followed in several countries of the world;

- Second, in other countries 'We face as a global public health challenge, as the number of deaths increases because the of baby boom.' This could well be a true phenomenon but the interpretation is not negative. See that some argue for active family planning to elicit a baby boom, thus wanted. Moreover, at the same time force of mortality could well be going down, thus it is not a public health problem but a late effect of an earlier societal behaviour;

- Third, for specific countries;' Life expectancy increases, population ageing accelerates, indicating that a global public health challenge is successfully dealt with.' For example, life expectancy in Russia dropped because there was a massive alcohol problem. Now this is partially dealt with, the population is ageing again, which is positive.

In conclusion, disaggregation of the impact of ageing on death is a research priority and can best be dealt with when population size, age structure and mortality rates are dealt with in all countries/regions separately, not by pooling what cannot be pooled.

Reviewer #3: I really enjoyed reading 'Impact of population ageing on global deaths, 1990-2017' manuscript. 

The article aimed to provide a novel robust method to evaluate the global impact of population ageing across a lengthy time period, which it has done successfully in my opinion.

The method showed in the manuscript is simple and robust as the authors claimed. The supplementary method is easy follow and it is sufficient for other researchers to reproduce. It'd be nice if the authors could expand the supplementary method to a tutorial with an actual example.

The manuscript is well organised and written clearly enough to be accessible to non-specialists. However, as a non native speakers I did not check any grammars.

More explanation of the method would help make the manuscript more complete without readers have to go to the supplementary. Word limits could be an issue but I think authors could shorten the discussion without losing anything important.

Reviewer #4: Population ageing is obviously associated with increased rates of mortality. In this paper, the authors use data from the Global Burden of Disease to estimate the magnitude of this over the period 1990-2017. To do this, the authors use a novel statistical approach they recently developed that can decompose death differences between populations based on population size, age structure of the population, and age-specific mortality rate. The authors found wide differences across income levels, countries and causes of deaths. 

Overall, I thought the paper was interesting and addressed an important topic. I have a few comments that I hope might improve the paper:

1. The authors could potentially report estimates for the increase in yearly deaths attributable to each year increase in a population's average age (controlling for income, country and other factors). Similar estimates could be given for increases in average income and changes in sex ratios. Such figures may be of interest to researchers in ageing and population planning. 

2. The authors currently give breakdowns of mortality attributed to individual diseases by sex. It could potentially be helpful to researchers of specific diseases to also have estimates of a disease's contribution to mortality associated with population ageing independent of sex. While this could be difficult given differences in sex ratios across countries and, as the authors point out, with ageing, I wonder if the authors could use their decomposition approach to provide this. 

3. In the section "Population Ageing", it could be helpful if the authors report the magnitude of ageing according to the income groupings (high, upper middle, lower middle, and low) that the authors use elsewhere in the results section. 

4. The authors report negative values in low income countries for the proportion of death increases attributable to population ageing. They explain: "The negative percentages in low-income countries reflect decreases in the proportion of old age groups" (p. 10). I found this slightly confusing - does this suggest that low-income countries' populations were negatively ageing (getting younger)? This point could be clarified in more detail.

5. Given the novelty of their statistical method, I wonder if sensitivity analyses could provide converging evidence for their findings (e.g., giving very brief examples of countries that differ in age but with similar incomes and reporting the differences in mortality etc.).

6. In the discussion, the authors discuss the tension between population ageing and mortality reduction from specific disease through prevention strategies and improved healthcare. This seems somewhat of a paradox to me - if mortality from a disease is reduced, I would have thought that would lead to further ageing, which, in turn, could eventually lead to mortality from other causes. Perhaps I am missing something, but maybe the authors could clarify this point and discuss this issue. 

The validity of the findings rests on a statistical approach that the authors themselves developed and published recently (Cheng et al., 2019, PLoS One). I do not have expertise in these statistics to evaluate this part of the paper. I would suggest that the editors obtain a review from a statistician in the area if they have not already.

[LINK]

---

## [Decision Letter · Decision Letter 1]

20 Apr 2020

Dear Dr. Hu,

Thank you very much for re-submitting your manuscript "Population ageing and mortality during 1990-2017: a global decomposition analysis" (PMEDICINE-D-19-03747R1) for review by PLOS Medicine.

I have discussed the paper with my colleagues and the academic editor and it was also seen again by three reviewers. I am pleased to say that provided the remaining editorial and production issues are dealt with we are planning to accept the paper for publication in the journal.

[LINK]

We look forward to receiving the revised manuscript by Apr 27 2020 11:59PM. 

Sincerely,

Thomas McBride, PhD

Senior Editor 

PLOS Medicine

plosmedicine.org

Requests from Editors:

1- Abstract Background: “older people” or “older adults” rather than “old people”.

2- Abstract Methods and Findings and going forward, please consider using “high- and middle-income countries” and “low-income countries” rather than “countries with middle to high income levels”, unless these categories in your study do not line up with the WHO categories. If the latter is true, please make the distinction clear.

3- Abstract Methods and Findings, line 37: perhaps clearer to say “Compared to 1990, 12 million additional global deaths in 2017 were attributed to population ageing, corresponding to 27.9% of total global deaths.” (If my rewording is accurate).

4- Line 48: rather than “about 50%”, please state the absolute numbers for the numerator and denominator.

5- Starting in the Abstract and throughout the manuscript, please avoid causal language (e.g., “Population ageing caused increases in the number of deaths in countries with middle to high income levels but not for countries with low income”) and use “attributable for” or similar.

6- Thank you for adding an Author Summary. Point number 7 (“The reduction in number of deaths from 1990 to 2017 from mortality change exceeds the increase of deaths caused by population ageing for the whole world and in about 50% of countries where population ageing was associated with increased death burden.”) is a bit confusing. It would be appropriate to split into two sentences to clarify.

7- Please qualify the first sentence of the last paragraph of the Introduction with “to our knowledge” or similar.

8- Line 167: please delete “robust”.

9- Line 189: please delete “new”.

10- Results section: thank you for including the absolute numbers that correspond to percentages in the tables. Please also include absolute numbers when citing the percentages in the text.

11- Line 386: if you describe the decomposition method as “robust”, please explain here what makes it robust.

12- Again on lines 400, 407, and 426, please either describe how this method is robust or delete “robust”.

Comments from Reviewers:

Reviewer #1: The authors have addressed my concerns and I now recommend publication

Peter Flom

Reviewer #2: The authors have taken the comments seriously which resulted in a more cautious interpretation of their efforts.

Reviewer #4: The authors have addressed my comments.

[LINK]

---

## [Editor Report · Decision Letter 2]

13 May 2020

Dear Dr. Hu, 

On behalf of my colleagues and the academic editor, Dr. Sanjay Basu, I am delighted to inform you that your manuscript entitled "Population ageing and mortality during 1990-2017: a global decomposition analysis" (PMEDICINE-D-19-03747R2) has been accepted for publication in PLOS Medicine. 

PRODUCTION PROCESS

PRESS

PROFILE INFORMATION

Thank you again for submitting the manuscript to PLOS Medicine. We look forward to publishing it. 

Best wishes, 

Thomas McBride, PhD

Senior Editor 

PLOS Medicine

plosmedicine.org